# Microstructural Characterization and Magnetic, Dielectric, and Transport Properties of Hydrothermal La_2_FeCrO_6_ Double Perovskites

**DOI:** 10.3390/nano13243132

**Published:** 2023-12-13

**Authors:** Kang Yi, Zhiwei Wu, Qingkai Tang, Jiayuan Gu, Jie Ding, Liangdong Chen, Xinhua Zhu

**Affiliations:** National Laboratory of Solid State Microstructures, School of Physics, Nanjing University, Nanjing 210093, China; 15255458807@163.com (K.Y.); 15062861103@163.com (Z.W.); tangqingkai@126.com (Q.T.); jiayuangu96@163.com (J.G.); dingjiewh123@163.com (J.D.); chenliangd198@163.com (L.C.)

**Keywords:** double perovskites, La_2_FeCrO_6_ oxides, hydrothermal process, magnetic properties, electrical transport properties, microstructural characterization

## Abstract

Double perovskite La_2_FeCrO_6_ (LFCO) powders were synthesized via the hydrothermal method, which crystallized in an orthorhombic (*Pnma*) structure and exhibited a spherical morphology with an average particle size of 900 nm. Fourier transform infrared spectroscopy demonstrated the presence of fingerprints of vibrational modes of [FeO_6_] and [CrO_6_] octahedra in the powders. The XPS spectra revealed dual oxide states of Fe (Fe^2+^/Fe^3+^) and Cr (Cr^3+^/Cr^4+^) elements, and the oxygen element appeared as lattice oxygen and defect oxygen, respectively. The LFCO powders exhibited weak ferromagnetic behavior at 5 K with a Curie temperature of 200 K. Their saturation magnetization and coercive field were measured as 0.31 μ_B_/f.u. and 8.0 kOe, respectively. The Griffiths phase was observed between 200 K and 223 K. A butterfly-like magnetoresistance (MR)–magnetic field (H) curve was observed in the LFCO ceramics at 5 K with an MR (5 K, 6 T) value of −4.07%. The temperature dependence of resistivity of the LFCO ceramics demonstrated their semiconducting nature. Electrical transport data were fitted by different conduction models. The dielectric behaviors of the LFCO ceramics exhibited a strong frequency dispersion, and a dielectric abnormality was observed around 260 K. That was ascribed to the jumping of electrons trapped at shallow levels created by oxygen vacancies. The dielectric loss showed relaxation behavior between 160 K and 260 K, which was attributed to the singly ionized oxygen vacancies.

## 1. Introduction

Perovskite transitional metal (TM) oxide nanostructures have gained a lot of attention due to their wide spectrum of intriguing properties [1,2,3] and promising technical applications [4]. Recently, nanostructures of double perovskites (DPs) A_2_B′B″O_6_ with hybrid 3*d*, 4*d*, or 5*d* TM ions at the B-site have attracted much attention because of their potential applications in the fields of microelectronic devices such as magnetic memories [5], magnetic tunnel junctions [6], and spin filtering devices [7]. In such a system, the interplay between the B′ (localized 3*d*-block TM ions) and B″ ions (delocalized 4*d* or 5*d*-block TM ions) provides more compositional flexibility to generate fascinating multifunctionalities [8,9]. Therefore, under the hybrid 3*d*- with 4*d*- or 5*d*-block TM ions at the B-site, highly spontaneous ordering can be achieved at the B-site because of their sizable differences in chemical valence states and the ionic radii between B′ and B″ TM ions. For example, structural ordered SrFeMoO_6_ [5], Sr_2_FeReO_6_ [10], and Sr_2_CrReO_6_ [11] bulk DP oxides have been successfully synthesized. However, in the special case of two 3*d*-block TM ions positioned at the B′ and B″ sites, the synthesis of the 3*d* (B′)–3*d* (B″) ordered DP oxides is much more difficult owing to their similar ionic radii [12,13].

As a representative 3*d*–3*d* DP oxide, La_2_FeCrO_6_ (LFCO) has drawn considerable attention since it provides a model system for testing the propensity for ferromagnetism, given this combination and configuration of Cr^3+^ and Fe^3+^ ions, and the general infallibility of the Goodenough–Kanamori (GK) rules [14,15]. In the past decade, several works have been conducted in the LFCO oxide system from theoretical and experimental aspects. Ueda et al. [16] first realized the ferromagnetic (FM) ordering in the artificial LaCrO_3_–LaFeO_3_ superlattices, which were fabricated by laser molecular beam epitaxy via alternative depositions of Cr^3+^ and Fe^3+^ ions on the (111) plane. The magnetic coupling between Cr^3+^ and Fe^3+^ ions was confirmed to be FM in terms of superexchange [14,15]. Unfortunately, the saturated magnetization, *M*_S_, was measured to be 3.0 μ_B_/f.u., which is much lower than the theoretical one (*M*_S_~7.0 μ_B_/f.u.). The Rietveld refinements on the neutron diffraction data indicated that the LFCO compound crystallized in an orthorhombic perovkite structure (*Pbnm*) with a random positioning of the Fe and Cr cations at the B-sited sublattices, displaying antiferromagnetic (AFM) behavior at 265 K [17]. That is in agreement with the AFM coupling expected from the linear *d*^3^(Cr^3+^)–*d*^3^(Cr^3+^) and *d*^5^(Fe^3+^)–*d*^5^(Fe^3+^) superexchange (SE) interactions [18]. According to the GK rules, the FM coupling between the Fe^3+^ and Cr^3+^ ions in the LFCO DP oxide with a rock-salt ordering is expected due to the SE interaction via the Fe(*d*^5^)–O–Cr(*d*^3^) magnetic path [14,15]. Recently, theoretical calculations have demonstrated a ferrimagnetic (FiM) ground state in an ordered LFCO DP oxide with Cr^3+^ and Fe^3+^ ions coupled antiferromagnetically [19]. This conclusion was experimentally verified in the well-ordered epitaxial LFCO thin films with a B-site ordering degree as high as 90% [20], and the film had an *M*_S_ value of 2.0 μ_B_/f.u. at 5 K, which was in accordance with the Pickett’s model [21], but the KG rules were broken. In the bulk LFCO samples, only the AFM behavior was observed, and the disappearance of the FM order was ascribed to the random positioning of the Fe^3+^ and Cr^3+^ ions at the B-sited sublattices [22]. As described above, different magnetic behaviors have been reported in the LFCO oxide artificial superlattice, thin films, and bulk counterparts, which are believed to be closely related to the diverse couplings of the Fe–O–Fe, Cr–O–Cr, and Fe–O–Cr bonds [16]. To date, although the structure and physical properties of the LFCO oxide system are widely studied, there is still much controversy in their magnetic behaviors because the magnetic properties of LFCO DP oxides are very susceptible to the formation of oxygen vacancies ($V_{\ddot{O}}$) and anti-site defects (ASDs) and to the volatilization of La during their high-temperature synthesis, which favors the formation of different chemical oxide states of Fe and Cr TM ions. A competition between the AFM and FM interactions via the different magnetic paths can result in different magnetizations in the LFCO oxides [23]. Thus, in order to explore the physical mechanisms behind these anomalous magnetic behaviors in LFCO DP oxides, more systematic investigations are highly required. Recently, Sun et al. [24] theoretically investigated the structural and magnetic properties of R_2_CrFeO_6_ (R = rare earth elements) DP oxides. They found an antiparallel alignment of the spins of the Fe and Cr TM ions existing in the R_2_CrFeO_6_ oxides, and such AFM interaction resulted in Ferrimagnetism (FiM). Since the FiM state has the lowest energy in the monoclinic *P*2_1_/*n* phase, it could be regarded as a ground state. They also found that by increasing the *R* radius, the energy difference between the *P*2_1_*/n* and *R*-3 phases were reduced. The theoretical calculations indicate that the magnetic properties of R_2_CrFeO_6_ DP oxides are not only dependent upon the Cr–O–Fe bond angles and tilting angles but also upon the different constituents of the material [24]. An insulating ferrimagnet LFCO with antiparallelly aligned *S* = 3/2 (Cr^3+^) and *S* = 5/2 (Fe^3+^) ions is predicted through different first-principles approaches [19]. Similarly, the magnetic orders in the (LaFeO_3_)_n_–(LaCrO_3_)_n_ superlattices (denoted as SL(n)) were investigated by Monte Carlo simulations, and the results were compared with those from the LaFe_0.5_Cr_0.5_O_3_ bulk counterpart [25]. It is also noticed that both FM and FiM behaviors appeared in the SL(1) and SL(3), whereas two different AFM orders occurred in the SL(2) and SL(4), respectively. The present results not only match well with the experimental data in the SL(1) but also demonstrate some novel ordered phases in the SLs constructed with other periods. However, the magnetic transport properties of the LFCO oxide system have not been reported yet despite the fact that they play important roles in the applications of spintronic devices.

In view of the above facts, in the present work, LFCO oxides were synthesized by the hydrothermal method. This process is characterized as a powerful method for the synthesis of perovskite oxide powders with controllable sizes and morphologies via the modifications of the hydrothermal processing parameters (e.g., precursor types, hydrothermal reaction temperature and time, pH value, and the type and concentration of the used mineralizers). The structural, magnetic, and dielectric properties of the hydrothermal LFCO DP oxides and their electrical and magnetic transport properties were comprehensively studied to better understand the relationships between the microstructure and physical properties of the LFCO oxides.

## 2. Materials and Methods

### 2.1. Synthesis of LFCO Powders

Hydrothermal process was used to synthesize the LFCO powders, which was carried out in a Teflon-lined stainless steel autoclave. The filling capacity of the autoclave was 80%. First, solutions of La(NO_3_)_3_·6H_2_O, Fe(NO_3_)_3_·9H_2_O, and Cr(NO_3_)_3_·9H_2_O were prepared with 0.5 M concentration, and the molar ratio of La:Fe:Cr was kept as 2:1:1. For a typical synthesis of the LFCO powders, 10 mL suspension with Cr(NO_3_)_3_·9H_2_O and Fe(NO_3_)_3_·9H_2_O was made with the addition of 2 g KOH. Then, 10 mL La(NO_3_)_3_ solution was put into the above suspension, followed by the addition of another 8 g KOH (used as a mineralizer) under continuous stirring for 30 min at room temperature (RT) to form a mixed solution. The above mixed solution was transferred into hydrothermal autoclaves, which were kept at 433 K for 4 h and then cooled naturally to RT in air. Deionized water was used to wash the resulting powders, which were filtered and dried at 353 K for 12 h in an oven. Finally, the obtained brown powders were further post-annealed at 1473 K for 12 h in air in a tube furnace.

### 2.2. Microstructural Characterization and Physical Properties

All of the post-annealed LFCO powders were characterized by X-ray diffraction (XRD) at RT by using a SIEMENS D5000 diffractometer (Simens, Berlin, Germany) under Cu Kα radiation (λ = 1.54056 Å). XRD data were collected in a step-scan mode with 2*θ* in the range of 20° to 80°. The step size was 0.02°/s, and the collecting time for each step was 10 s. Structural parameters of the LFCO powders were extracted from Rietveld refinements on the powder XRD data by using the general structure analysis system (GSAS) software (GSAS-II) [26]. Fourier transform infrared (FTIR) spectrum of the LFCO powders was recorded in the wavenumber range of 400 cm^−1^ to 1600 cm^−1^ by using a PerkinElmer FT-IR 65 spectrometer. The morphology and chemical compositions of the tested samples were examined using a scanning electron microscope (SEM, FEI QUANTA 650, Hillsboro, OR, USA) with attached energy-dispersive X-ray spectroscopy (EDS). The EDS data were acquired in a mapping mode. X-ray photoelectron spectroscopy (XPS) was utilized to determine the chemical binding energies (BEs) and oxide states of the constituent elements in the LFCO powders. The La 3*d*, Fe 2*p*, Cr 2*p*, and O 1*s* XPS spectra were collected by using a PHI 5000 spectrometer (Versa Probe, ULVAC-PHI, Kanagawa, Japan) under Al Kα radiation at 1486.60 eV as X-ray source. All of the collected XPS spectra were calibrated by C 1*s* core-level positioned at a BE value of 284.60 eV. Dielectric measurements of the LFCO ceramics were carried out by using an Agilent impedance analyzer (Model, Agilent 4192 A, Agilent technologies, Santa Clara, CA, USA) operated in a frequency range of 10^2^ Hz to 10^6^ Hz and a temperature range of 173–373 K controlled by a controller (DMS-2000, Partulab Technology, Wuhan, China). The D.C. magnetization (M) data were measured using a SQUID magnetometer (MPMS3, Quantum Design, San Diego, CA, USA). The M-T curves were recorded in the ZFC (zero-field cooling) and FC (field cooling) protocols under a magnetic field of 500 Oe over a temperature range of 2 K to 300 K. The *M*-*H* hysteresis loops were measured at 5 K and 300 K, respectively, within the magnetic field of ±6 T. All of the magnetic measurements were performed with the powder samples installed inside a Teflon capsule. A standard four-probe method was used to measure the resistivity (*ρ*) of the LFCO ceramics over a temperature range of 2 K to 800 K without applying a magnetic field. The magnetic field dependence of *ρ* for the LFCO ceramics was also measured from −6 to 6 T at 5 K and 300 K, respectively, from which the plot of magnetoresistance (MR) versus magnetic field (H) was extracted.

## 3. Results and Discussion

### 3.1. Microstructural Characterization

The XRD pattern of the LFCO powders measured at RT is shown in Figure 1. All of the XRD peaks can be indexed in an orthorhombic perovskite structure with a space group of *Pnma* (JCPDS file, No. 89-0478), indicating that the LFCO powders crystallize in an orthorhombic lattice symmetry. The profile of the Rietveld refinements on the powder XRD data and the allowed Bragg reflections are also demonstrated in Figure 1. Three reliability factors (*R*_p_, *R*_wp_, and *χ*^2^) were utilized to assess the fitting quality between the experimental XRD data and the theoretical ones, which were determined as *R*_wp_ = 6.03%, *R*_p_ = 5.98%, and *χ*^2^ = 2.32.

The small value of *χ*^2^ indicated a good fittingness. The refined structural parameters at RT are tabulated in Appendix A, matching well with the data reported previously [27,28]. The Goldschmidt’s tolerance factor (*t*) of the LFCO powders was also calculated to be 0.961, indicating that an orthorhombic crystal structure was preferred for the present LFCO powder [29]. It is also noticed that any XRD peak representing the B-site cationic ordering does not appear in Figure 1, suggesting that the rock-salt type ordering of the Fe and Cr ions at B-sited sublattices is not realized due to their similar ionic radii and the same chemical valence states [30]. The Scherrer formula described by Equation (1) was used to determine the average crystallite size (*D*) of the LFCO powders [31]:(1)$$D=\frac{0.9\lambda }{\beta cos\theta_{B}}$$
where *λ* is the X-ray wavelength of the Cu Kα radiation (*λ* = 1.54056 Å), *β* is the full width at half maximum (HWHM) of the (200)/(121) diffraction peak (in radians), and *θ*_B_ is the Bragg diffraction angle. The average crystallite size, *D*, was determined to be 41.0 nm. 

The typical low-magnification SEM image of the LFCO powders is displayed in Figure 2a, and Figure 2b presents the histogram of the LFCO particle size distribution. It was noticed that the LFCO particles had a spherical morphology and their average particle size was determined to be 0.90 μm by fitting the particle size distribution in Figure 2b. Figure 2c displays an EDS spectrum acquired from the LFCO powders in a mapping mode, illustrating the EDS signals of the constituent elements, as expected. There are no other elements in the LFCO powders. The atomic molar ratio of La:Fe:Cr was determined to be 2:0.94:0.93 from the quantitative EDS data, which approached the nominal chemical compositions of the powders.

FTIR spectroscopy was used to identify the local structures of the hydrothermal LFCO powders. Figure 3 shows the FTIR spectrum obtained from the powders, where two absorption bands positioned at 437 cm^−1^ and 603 cm^−1^ are clearly observed. The weak absorption band located at 437 cm^−1^ is attributed to the bending vibrations of the Fe–O bonds within the FeO_6_ octahedron [32], while the intense absorption band positioned at 603 cm^−1^ is a result of the stretching vibration of the Cr-O bonds within the CrO_6_ octahedra [33]. Thus, the octahedral coordination of Fe and Cr ions with oxygen ions (O^2−^) in the powders is confirmed by the FTIR spectrum.

The chemical valence states of the constituent elements (La, Fe, Cr, and O) in the LFCO powders were verified by the XPS spectra at RT. Figure 4a exhibits a wide scanning XPS spectrum for the LFCO powders over the scanning energy from 200 to 1000 eV, where the La 3*d*, Fe 2*p*, Cr 2*p*, and O 1*s* core-level XPS peaks are clearly observed. In addition, the C 1*s* core-level XPS peak is also observed at 284.60 eV, which is attributed to the conductive carbon tape employed in the XPS measurements. High-resolution XPS spectra of the La 3*d*, Fe 2*p*, Cr 2*p*, and O 1*s* core levels are demonstrated in Figure 4b–e, respectively. In Figure 4b, the La 3*d* core-level XPS spectrum is deconvoluted into two doublets positioned in the 831–840 eV and 850–858 eV regions, respectively. The former doublet corresponds to the La 3*d*_5/2_ level, and the latter one corresponds to the La 3*d*_3/2_ level. In addition, the satellite peaks of the La 3*d*_5/2_ and La 3*d*_3/2_ XPS peaks appeared at 838.43 eV and 855.32 eV, respectively. These two satellite peaks are ascribed to the electron transferring from the oxygen valence band to the empty La 4*f* level [34]. The difference (Δ) between the BE values of the La 3*d*_5/2_ (834.47 eV) and La 3*d*_3/2_ (851.35 eV) XPS peaks was measured to be 16.88 eV, which corresponds to the spin–orbit coupling (SOC) of the La element. The Δ value of the La element as well as the BE values of the La 3*d*_5/2_ and La 3*d*_3/2_ XPS peaks confirm the presence of La^3+^ ions in the LFCO powder sample [35]. Figure 4c depicts the Fe 2*p*_3/2_ core-level XPS spectrum over the scanning energy range of 706–715 eV, where two characteristic peaks are observed at 710.15 eV and 711.40 eV, respectively. They can be assigned to Fe^2+^ and Fe^3+^ ions, respectively [36]. The percentage molar ratio of Fe^2+^ to Fe^3+^ species was extracted from the peak fitting of the Fe 2*p*_3/2_ XPS spectrum, which was 18%:82%. Thus, the effective oxide state of the Fe ion was +2.82. Figure 4d displays the Cr 2*p*_3/2_ XPS spectrum over the energy region of 574–582 eV. Similarly, the Cr 2*p*_3/2_ XPS peak is also deconvoluted into two peaks located at BE values of 576.25 eV and 578.90 eV, respectively. They are assigned to the Cr^3+^ and Cr^4+^ species, respectively [37,38]. The percentage molar ratio of the Cr^3+^ to Cr^4+^ species was calculated as 74%:26%. Therefore, the effective oxide state of the Cr ions was +3.26. In Figure 4e, the O 1*s* XPS spectrum in the scanning energy region of 526–536 eV exhibits an asymmetric feature, indicating more than one kind of oxygen species in the powders. Previously, three kinds of oxygen species named as lattice oxygen (denoted as O_α_), defect oxygen (represented as O_β_), and surface adsorbed oxygen (designated as O_γ_) were reported to appear in the energy regions of 529–530 eV, 530–532 eV, and 533–534 eV, respectively [39,40]. Here, only two types of oxygen species appear in the present O 1*s* XPS spectrum, which are lattice oxygen (O_α_) with a BE value of 529.42 eV and defect oxygen (O_β_) with a BE value of 531.70 eV, respectively. The percentage molar ratio of the O_α_ to O_β_ species was estimated to be 46%:54%. This result indicates a higher molar ratio of oxygen defects in the LFCO powders. The species, peak positions, and percentage molar ratios obtained from the peak fittings of the Fe 2*p*_3/2_, Cr 2*p*_3/2_, and O 1*s* XPS spectra are summarized in Appendix A, where the effective chemical oxidation states of the Fe and Cr ions are also presented.

### 3.2. Magnetic Properties

Figure 5a displays the *M*-*H* hysteresis loops of the LFCO powders recorded at 5 K and 300 K, respectively. It is noticed that the *M*-*H* curve of the LFCO powders exhibits a linear-like behavior at 300 K with almost zero remnant magnetization and coercivity, indicating paramagnetic-like behavior, while at 5 K, an open hysteresis loop is clearly observed, which is saturated at a particular magnetic field, *H*_F_ = 40 kOe. Beyond 40 kOe, the magnetization increases linearly with the magnetic field, indicating AFM behavior in the LFCO powders. That is ascribed to the SE interactions via the Fe^3+^–O–Fe^3+^ and Cr^3+^–O–Cr^3+^ magnetic paths. In addition, the observed small open hysteresis loop indicates a weak ferromagnetism, which is a result of the double-exchange interactions via the Fe^3+^–O–Cr^3+^, Fe^3+^–O–Cr^4+^, Fe^2+^–O–Cr^3+^, and Fe^2+^–O–Cr^4+^ magnetic paths. The magnetic parameters such as the remanent magnetization (*M*_r_) and coercive field (*H*_c_), could be obtained from the *M*-*H* hysteresis loop recorded at 5 K, which were *M*_r_ = 0.23 emu/g (or 0.02 μ_B_/f.u.) and *H*_c_ = 8.0 kOe, respectively. It is noticed that beyond the *H*_F_, the *M*-*H* curve exhibits a linear relationship; thus, the measured *M-H* curve is composed of two parts: one is from the saturated hysteresis loop at *H*_F_, and another one is from the AFM magnetization following a linear relationship with the magnetic field. Therefore, the *M*(*H*) at the high-field region can be expressed by Equation (2) [41]:(2)$$M(H)=\chi_{AF}H+M_{S}$$
where the $\chi_{AF}H$ part is obtained from the AFM magnetization and *M*_S_ is the saturation magnetization of the weak ferromagnetism. To evaluate the *M*_S_ value of the samples, the law of approach to saturation is utilized, according to Equation (3) [42]: (3)$$M=M_{S}\left(1-\frac{A}{\sqrt{H}}-\frac{B}{H}\right)$$
where *A* and *B* are constants, which are related to the micro-stress and magneto-crystalline anisotropy of the samples, respectively. Figure 5b displays the *M*(*H*) plotted as a function of $\frac{1}{\sqrt{H}}$, from which the *M*_S_ of the LFCO powders can be extracted as $\frac{1}{\sqrt{H}}$ approaching zero (or *H*→∞). Thus, the extracted *M*_S_ value of the LFCO powders from Equation (3) was 3.64 emu/g (or 0.31 μ_B_/f.u.) at 5 K. This value was higher than the previously reported *M*_S_ values for the hydrothermal LaFe_0.5_Cr_0.5_O_3_ powders (*M*_S_ = 0.21 μ_B_/f.u.) [43], LFCO nanoparticles (*M*_S_ = 0.046 μ_B_/f.u.) synthesized via the citrate auto-combustion technique [44], and LaFe_0.5_Cr_0.5_O_3_ ceramics (*M*_S_ = 0.04 μ_B_/f.u.) prepared by the solid-state reaction method [17]. However, the present *M*_S_ value was still much smaller than the theoretical value (*M*_S_ = 4.0 μ_B_/f.u.) predicted for the LaFe_0.5_Cr_0.5_O_3_ compound with an atomic order of Fe^3+^(*d*^5^)-O-Cr^3+^(*d*^3^) under high-spin states [45] or the *M*_S_ = 2.0 μ_B_/f.u. reported for the LFCO thin films with a B-site ordering degree of ~90% [20]. Based on the theoretical calculations of the local spin density, Pickett et al. reported that the FiM ground state in the LCFO compound with an *M*_S_ of 2 μ_B_/f.u. exhibited much more stability than that in the FM ones with an *M*_S_ of ∼7.0 μ_B_/f.u. [21]. It is known that the magnetic properties of the LFCO samples are influenced not only by the magnetic coupling manners between the Fe^3+^ and Cr^3+^ ions at the B-sited sublattices (e.g., long range FM or AFM order), but also by the ordering degree of the Fe^3+^ and Cr^3+^ ions at B sublattices (e.g., fully ordering, partial ordering, or complete disordering) due to their different *d*-orbitals and occupied states (e.g., *t*_2g_ or *e*_g_ orbitals; empty, half-filled, or fully filled orbital states) and different magnetic moments. The present XRD data reveal the almost random positioning of the Fe^3+^ and Cr^3+^ ions at B-sited sublattices, which destroys the long-range alignments of magnetic moments. That is the reason why the experimental *M*_S_ value is much smaller than the theoretical *M*_S_ value (2.14 μ_B_/f.u.) of the LFCO powders predicted based on an intuitive ionic model. The concentration of ASDs in the LFCO powders was evaluated to be 42.8%, and the corresponding B-site ordering degree (η) in the LFCO powders was determined to be 14.4%. The details for the calculations of the ASD concentration and B-site ordering degree are described in Appendix A. The temperature dependence of the *M*_ZFC_ and *M*_FC_ curves is shown in Figure 5c. The two M-T curves demonstrate a typical ferrimagnetic (FiM) transition around 200 K, which is a field-independent magnetic phase transition at *T*_C_ = 200 K, as confirmed by the derivative dM/dT curves for both the *M*_ZFC_ and *M*_FC_, and it is plotted as an inset of Figure 5c. Such a transition temperature is smaller than that reported for the bulk samples (*T*_C_ = 265 K) [17]. The difference (Δ*T*_C_ = 65 K) can be ascribed to the finite size effect and/or surface strain effect of nanopowders [46]. Upon cooling, the *M*_ZFC_ curve varied very smoothly, and the paramagnetic (PM)-to-FiM transition behavior was very weak, and a much broad phase transition peak appeared. The blocking temperature (*T*_B_) associated with this broad phase transition can be determined to be 113 K from the minimum position of the first derivative d*M*_ZFC_/d*T* curve (see inset in Figure 5c). The low-temperature magnetic transition from the FiM to AFM phases appeared around *T*_N_ = 10 K. A large irreversibility (Δ*M*, defined as (*M*_FC_–*M*_ZFC_)/*M*_FC_) was observed between the ZFC and FC curves, and the Δ*M* value reached ~71% at 50 K. That was ascribed to the fast increase in the predominant alignments of the spin orientations under 500 Oe. It was also noticed that the *M*_ZFC_ and *M*_FC_ curves did not merge together at *T*_C_, and the irreversibility was still maintained well above *T*_C_. That indicated that the true paramagnetic behavior was not realized immediately above *T*_C_. It was noticed that the magnetic moments (*M*) of the LFCO powders were much smaller (in the range of 0.01–0.05 emu/g) within the investigated temperature range (2–300 K), which is attributed to the almost random positioning of the Fe^3+^ and Cr^3+^ ions at the B-sited sublattices and/or oxygen vacancies ($V_{\ddot{O}}$). Figure 5d shows the temperature-dependent inverse of the susceptibility (*χ*^−1^), where *χ*^−1^ varies linearly with the temperature in the range of 225–280 K, indicating PM behavior. Thus, *χ*^−1^(*T*) follows the Curie–Weiss (CW) law, as expressed by Equation (4):*χ*^−1^(*T*) = (*T* − *θ*_p_)/*C*(4)
where *C* denotes the CW parameter and *θ*_P_ represents the PM Curie temperature, which are tabulated in Appendix A. The negative *θ*_P_ (*θ*_P_ = −441 K) suggests the predominant AFM interaction in the LFCO powders, which accords well with the AFM behavior reflected by the M-H loop at 5 K. As the temperature further decreases below 223 K, *χ*^−1^ decreases smoothly in a quasi-linear manner at an extended temperature range. From the CW parameter, *C*, the effective magnetic moments (*μ*_eff_) in the PM phase can be calculated as 6.64 μ_B_/f.u. (@500 Oe), and the theoretical magnetic moment, *μ*_cal_ (per formula unit), of the La_2_${Fe}_{0.18}^{2+}{Fe}_{0.82}^{3+}{Cr}_{0.74}^{3+}{Cr}_{0.26}^{4+}$O_6_ can be calculated as 6.80 μ_B_/f.u. Details of the *μ*_eff_ and *μ*_cal_ calculations are described in Appendix A. It is noticed that the *μ*_cal_ of the LFCO powders is slightly larger than the *μ*_eff_ under a magnetic field of 500 Oe. This is attributed to the weak magnetic coupling between the Fe and Cr ions because of the high concentration of ASD content in the LFCO powders. The calculated *μ*_eff_ and *μ*_cal_ for the LFCO powders are tabulated in Appendix A. In Figure 5d, the *χ*^−1^ − *T* curve exhibits a severe downturn deviation from the CW law below the temperature of 223 K (which is denoted as the Griffiths temperature, $T_{C}^{G}=223K$), indicating the existence of a Griffiths phase (GP) in the LFCO powders. The GP phase lies between the magnetically ordered state and the completely disordered paramagnetic high-temperature regime [47]. In the GP region, *χ*^−1^ − *T* holds a power law described by Equation (5) [48,49]:(5)$$\chi^{-1}(T)\;\propto \;(T-T_{C}^{R}{)}^{1-\gamma }$$
where *γ* is an exponent, indicating the strength of GP, and $T_{C}^{R}$ is a random critical temperature. The $T_{C}^{R}$ can be obtained from *γ* = 0 in the CW regime [50] and is equivalent to the *θ*_P_. Thus, *γ* = 0 represents the PM state, and *γ* = 1 represents a GP phase. A nonlinear fitting curve of *χ*^−1^ in the temperature range between *T*_C_ and $T_{C}^{G}$ is presented in Figure 5d, from which the *γ* value is determined as 0.268, and the corresponding value of $T_{C}^{R}$ is 146.2 K. The small *γ* value indicates the weak Griffiths singularity in the LFCO powders.

### 3.3. Dielectric Properties

At RT, the dielectric properties of the LFCO ceramics measured as a function of the frequency is illustrated in Figure 6a. It is observed that both the dielectric constant (*ε*_r_) and dielectric loss (tan*δ*) fall continuously as the frequency increases, displaying a dielectric behavior with a strong frequency dispersion. Rapid decreases in *ε*_r_ and tan*δ* at low frequencies can be contributed to the onset of several polarization mechanisms (e.g., space charge and dipolar, ionic, and electronic polarizations). Under the applied electric field, electrons can hop across the grains and grain boundaries of the LFCO ceramics. Charge carriers can pile up at the grain boundaries due to their higher resistance, leading to a space charge polarization [50]. The dipolar polarization and interfacial polarization make significant contributions to the larger *ε*_r_ at low frequencies. In order to make these charge carriers move along grain boundaries, much energy is needed, leading to a high value of tan*δ*. At higher frequencies, only the electronic and ionic polarization mechanisms are capable of reacting to the applied electric field, leading to smaller values of *ε*_r_ and tan*δ*.

In addition, the charge carriers gathered at the grain boundaries are also reduced at higher frequencies. Thus, these confined charge carriers are scattered, resulting in smaller *ε*_r_ and tan*δ* values. Such dielectric dispersion can be well explained based on the Maxwell–Wagner relaxation model [51].

Under a series of frequencies, the *ε*_r_ and tan*δ* values of the LFCO ceramics were measured as a function of temperature, and the results are displayed in Figure 6b and Figure 6c, respectively. As shown in Figure 6b, the *ε*_r_ grows slowly in the low-temperature region below 200 K, whereas beyond 200 K, the *ε*_r_ value increases fast and reaches a maximum peak at a temperature of around 260 K, and then it decays following the CW law. In addition, a collapse of the *ε*_r_ peak value takes place as the frequency is increased, but the peak positions do not shift towards the high-temperature direction. That is much clearly observed in the *ε*_r_–*T* curves measured at low frequencies (1 kHz–40 kHz). This dielectric abnormality can be attributed to the jumping of electrons that are weakly bound to $V_{\ddot{O}}\;\;or\;\;V_{\overset{\cdot }{O}}$ vacancies. These trapped electrons can be excited into the conduction band of the LFCO ceramics under thermal excitation at ~260 K. The thermal energy (*E* = *k*_B_*T*) at 260 K is 0.0224 eV, close to the activation energy (*E*_a_ = 0.02 eV) of the jumping electrons. As the applied electric field frequency increases, the trapped electrons are unable to catch up with the field, leading to a smaller contribution to the *ε*_r_. Thus, a collapse of the *ε*_r_ value with an increasing frequency appears, as shown in Figure 6b. However, at higher frequencies (e.g., 400, 800, and 1000 kHz), the maximum peak of *ε*_r_ around 260 K disappeared; instead, a diffused phase transition occurred, which was mainly due to the almost random positioning of Fe and Cr cations at the B-sited sublattices in the LFCO ceramics with a high content of ASDs. The compositional fluctuations at the B-site can result in microscopic heterogeneity in the LFCO compound, leading to statistics of local Curie points around the average one. In addition, at higher frequencies, the contributions to the *ε*_r_ peak from the dipolar and interfacial polarizations become much less significant because the two polarization mechanisms are no longer operative to respond to the applied electric field. In Figure 6c, tan*δ* has a much smaller value in the low-temperature region below 260 K, and beyond 260 K, a fast increase in tan*δ* appears, especially for those measured at low frequencies. The tan*δ*–*T* curves in the local temperature between 180 K and 260 K is shown as an inset in Figure 6c, where the relaxation peaks for tan*δ* move towards q higher temperature with an increasing frequency from 1 kHz to 10 kHz, and the peak values also collapse simultaneously. This dielectric anomaly exhibiting the typical characteristics of dielectrics are associated with the oxygen vacancies ($V_{\ddot{O}}$) and the related defect dipoles [52]. In the LFCO compound with a high content of ASDs, the ASDs can appear in the manner of the Fe-on-Cr site (Fe_Cr_) or Cr-on-Fe site (Cr_Fe_), respectively. Due to their different chemical oxide states, ${Fe}_{Cr}^{\prime }$ (Fe^2+^ on Cr^3+^ site or Fe^3+^ on Cr^4+^ site) and ${Fe}_{Cr}^{\prime \prime }$ (Fe^2+^ on Cr^4+^ site) defects are readily formed at B-sites. The ${Fe}_{Cr}^{\prime }$ and ${Fe}_{Cr}^{\prime \prime }\;$defects can attract the $V_{\ddot{O}}$ to form defect dipoles, namely ${{(Fe}_{Cr}^{\prime }-V_{\ddot{O}})}^{\cdot }\;$or ${Fe}_{Cr}^{\prime \prime }-V_{\ddot{O}}.$ These defect dipoles can change their orientations with respect to the jumping of O^2−^ into a vacant oxygen site of the oxygen octahedron or due to the hopping of electrons (among Fe^2+^/Fe^3+^ and/or Cr^3+^/Cr^4+^ pairs) that are weakly bound to $V_{\ddot{O}}\;or$ $V_{\overset{\cdot }{O}}$. At a low temperature below 200 K, the $V_{\ddot{O}}$ defects, ${{(Fe}_{Cr}^{\prime }-V_{\ddot{O}})}^{¨}\;$or ${Fe}_{Cr}^{\prime \prime }-V_{\ddot{O}}\;$defect dipoles, and the free electrons jumping between the Fe^2+^/Fe^3+^ and/or Cr^3+^/Cr^4+^ pairs, are bound at their respective defect sites. Thus, they make a small contribution to the *ε*_r_ and tan*δ* values at low temperatures [53]. However, a fast increase in the *ε*_r_ value appeared around 260 K at low frequencies, which was attributed to the $V_{\overset{\cdot }{O}}\;or$ $V_{\ddot{O}}$ defects in the LFCO ceramics, which were capable of moving across the whole sample, leading to the onsets of several polarization mechanisms (e.g., space charge and interfacial and/or defect dipolar polarizations) and affecting the dielectric relaxation behavior of the entire system [54]. In addition, the thermal excitation of electrons that are weakly bound to $V_{\ddot{O}}\;or\;V_{\ddot{O}}$ vacancies and their hopping between Fe^2+^/Fe^3+^ and/or Cr^3+^/Cr^4+^ pairs also contribute to the rapid increase in the *ε*_r_ value around 260 K at low frequencies. Beyond 260 K, the *ε*_r_ value decays following the CW law. As shown in Figure 6c, the sharp increase in tan*δ* at temperatures beyond 260 K can be ascribed to the rapid increase in the electrical conduction of the LFCO ceramics. To determine the *E*_a_ value associated with the dielectric relaxation behavior exhibited by tan*δ* between 160 K and 260 K, a plot of Ln*ω* versus 1000/*T* is used, as described by Equation (6):(6)$$\omega =\omega_{0}\exp\left(\frac{-E_{a}}{k_{B}T}\right)$$
where *ω* represents the angular frequency corresponding to the tan*δ* peak value, *ω*_0_ denotes the characteristic relaxation angular frequency at infinite temperature, and *k*_B_ and *T* have their normal meanings, as described in the textbook. When fitting the Ln*ω* vs. 1000/*T* curve with the Arrhenius law, a linear behavior between Ln*ω* and 1/*T* was observed, as shown in Figure 6d, from which the *E*_a_ value was evaluated to be 0.586 eV. It was reported that the *E*_a_ value for the singly ionized $V_{\overset{\cdot }{O}}$ in perovskite oxides is about 0.3–0.5 eV, while the value is ~1.0 eV for the doubly ionized $V_{\ddot{O}}$ vacancies [55]. The present *E_a_* value was 0.586 eV, close to that reported for the $V_{\overset{\cdot }{O}},$ implying that this dielectric relaxation is associated with the movement of singly ionized $V_{\overset{\cdot }{O}}$.

### 3.4. Electrical and Magnetic Transport Properties

Figure 7a shows the resistivity Ln*ρ*(*T*) of the LFCO ceramics measured with respect to the temperature without applying magnetic fields, where a continuous decrease in Ln*ρ*(*T*) within the investigated temperature range (2–800 K) is verified by the negative $\frac{dLn\rho }{dT}$. That reveals the semiconducting nature of the LFCO ceramics. Notably, around 20 K, the plot of dLn*ρ*/d*T* against *T* exhibits a valley, indicating a steep reduction in the resistivity Ln*ρ*(*T*) taking place around 20 K. This phenomenon could be ascribed to the phase transition from antiferromagnetic to ferrimagnetic phases as the temperature is increased. Such a phase transition could make the spin states fluctuate around 20 K, leading to a sharp increase in the charge carrier concentrations. Correspondingly, a steep decrease in the resistivity Ln*ρ*(*T*) appeared. To clarify the conduction mechanisms of the LFCO ceramic samples over 2–800 K, different conduction models such as Mott’s variable range hopping (VRH) model [56], the thermal activation model [57], and the small polaron hopping (SPH) model [58] are used to fit the resistivity data. In the high-temperature region of 350–683 K, the VRH model described by Equation (7) [56] was used to fit the resistivity data.
(7)$$\rho =\rho_{0}\exp{\left(\frac{T_{o}}{T}\right)}^{\frac{1}{4}}$$
where $\rho_{0}$ is a resistivity pre-factor, and *T*_o_ *=* 18/*k*_B_*N*(E_F_)*η*^3^, denoting the Mott characteristic temperature. Here *k*_B_, *N*(E_F_), and *η* stand for the Boltzmann’s constant, the density of electronic states at the Fermi level (E_F_), and the electron localization length, respectively. Figure 7b shows a linear fitting of the plot of the ln*ρ*(*T*) vs. (1000/*T*)^1/4^ curve in the temperature between 350 K and 683 K, which indicates that the electrical transport behavior of the LFCO ceramics in this temperature region is controlled by Mott’s VRH mechanism. The linear fitting of the ln*ρ*(*T*) vs. (1000/*T*)^1/4^ curve yields a *T*_o_ value of 5989.7 K, and the corresponding *N*(E_F_) value of 4.42 × 10^24^ eV^−1^ cm^−3^, where the *η* value for the small polarons was chosen as 1.99 Å at the scale of the average bong length of the <Fe-O> or the <Cr-O> bond. The VRH transport is typically observed in a system with a random potential. Such random potential is provided by the random positioning of Fe and Cr ions at B-sited sublattices in the present samples, as confirmed by the above XRD patterns and magnetic data. A slight deviation from the linear curve below 350 K in Figure 7b may be related to the magnetic scattering from the localized moments of Fe/Cr ions. In the temperature region between 275 K and 350 K, the plot of Ln*ρ* versus the inverse temperature (1000/*T*) shown in Figure 7c exhibits almost perfect linear behavior. Such kind of *T*-dependent resistivity is usually observed in semiconductors owing to thermal activation, which is described as by Equation (8) [57]:(8)$$\rho =\rho_{O}\exp(\frac{E_{a}}{k_{B}T})$$

The linear fitting gives out the *E*_a_ value of 11.43 meV, which is comparable to the dopant levels in normal semiconductors. In the temperatures below 275 K, the plot of the ln*ρ* vs. 1000/*T* curve becomes more deviated from a linear fitting. Instead, in the temperature between 15 K and 77 K, two linear fittings can be found for ln*ρ*/*T* vs. (1000/*T*), as illustrated in Figure 7d. That indicates that the electrical conduction is governed by the SPH mechanism in this temperature range, as described by Equation (9) [58]:(9)$$\rho =\rho_{O}T\exp(\frac{E_{a}}{k_{B}T})$$

The linear fitting of the ln(*ρ*/*T*) vs. (1000/*T*) curve shown in Figure 7d yields the *E*_a_ values for the small polarons hopping, which are 4.85 meV and 2.20 meV, respectively, in the two temperature regions, I (30–77 K) and II (15–30 K).

Magnetoresistance (MR) is described as a fractional change in the electrical resistance of the material when subjected to a magnetic field (*H*) at temperature (*T*), which is defined by Equation (10) [59]:(10)$$MR\left(T,H\right)=\left[\rho \left(T,H\right)-\rho \left(T,H_{peak}\right)\right]/\rho (T,H_{\mathrm{peak}})$$
where *H*_peak_ is the magnetic field at which the resistivity *ρ* reaches the maximum value. The MR dependent upon the field for the LFCO ceramics was measured at 5 K and 300 K, which is demonstrated in Figure 7e and Figure 7f, respectively. Negative MR behavior was observed at 5 K and 300 K. It is noticed that in Figure 7e, the MR-H plot at 5 K exhibits a typical butterfly-like shape, and the MR (5 K, 6 T) is measured to be −4.07%. Such MR value is higher than that reported for the semiconducting DP oxides such as Sr_2_CrReO_6_ ceramics with MR (4.2 K, 7 T) = −3.0% [60], Sr_2_CrHfO_6_ ceramics with MR (2 K, 7 T) = −2.73% [61], and Sr_2_Fe_0.5_Hf_1.5_O_6−δ_ ceramics with MR (2 K, 7 T) = −2.05% [62]. However, the butterfly-like MR-H curve disappeared at 300 K in Figure 7f, which was attributed to the absence of *M*-*H* hysteresis during the magnetic domain rotation under the excitation of the magnetic field. In addition, the MR value varied rapidly in a small magnetic field region but slowly in the high magnetic fields, exhibiting a similar response of the steep magnetization. That indicates that such negative MR behavior attributed to the magneto-tunneling effect in the LFCO ceramics.

## 4. Conclusions

In summary, we successfully synthesized the LFCO DP powders via the hydrothermal process and investigated their structural, magnetic, dielectric, and electrical/magnetic transport properties. The Rietveld refinement on the powder XRD pattern revealed that the LFCO powders crystallized in an orthorhombic distorted perovskite structure with the *Pnma* space group. The SEM images demonstrated the spherical morphology of the LFCO powders with an average particle size of 0.90 μm. The EDS data gave out the atomic ratio of La:Fe:Cr equal to 2:0.94:0.93. FTIR spectroscopy demonstrated the structural units of the [FeO_6_] and [CrO_6_] octahedra in the powders based on their fingerprints of vibrational modes. The XPS spectra verified the dual chemical valence states of the Fe (Fe^2+^/Fe^3+^) and Cr (Cr^3+^/Cr^4+^) elements as well as two kinds of oxygen species, namely lattice oxygen and defect oxygen. The LFCO powders exhibited weak ferromagnetic behavior at 5 K with *M*_S_ = 0.31 μ_B_/f.u. and *H*_c_ = 8.0 kOe, respectively. The *T*_C_ was determined to be 200 K, and a strong irreversibility between the *M*_ZFC_ and *M*_FC_ curves appeared around 295 K, which was attributed to the fast increase in the predominant alignments of the magnetic clusters under an external magnetic field. A Griffiths phase appeared in the temperature range between 200 K and 223 K. The LFCO ceramics exhibited butterfly-like *MR-H* behavior at 5 K, and the MR (5 K, 6 T) value was measured to be −4.07% due to the magneto-tunneling effect. The temperature-dependent resistivity data for the LFCO ceramics exhibited semiconducting behavior, and the resistivity data were well fitted by different conduction models in different temperature regions. The LFCO ceramics displayed dielectric behavior with a strong frequency dispersion, and the dielectric abnormality observed around 260 K was attributed to the jumping of electrons that were trapped in a shallow level created by oxygen vacancies. The activation energy associated with dielectric relaxation behavior, which was exhibited by the dielectric loss between 160 K and 260 K, was determined to be 0.586 eV, approaching the data reported for a singly ionized $V_{\overset{\cdot }{O}}$. The movements of singly ionized $V_{\overset{\cdot }{O}}\;$vacancies are responsible to such relaxor dielectric behavior.

## Figures and Tables

**Figure 1 nanomaterials-13-03132-f001:**
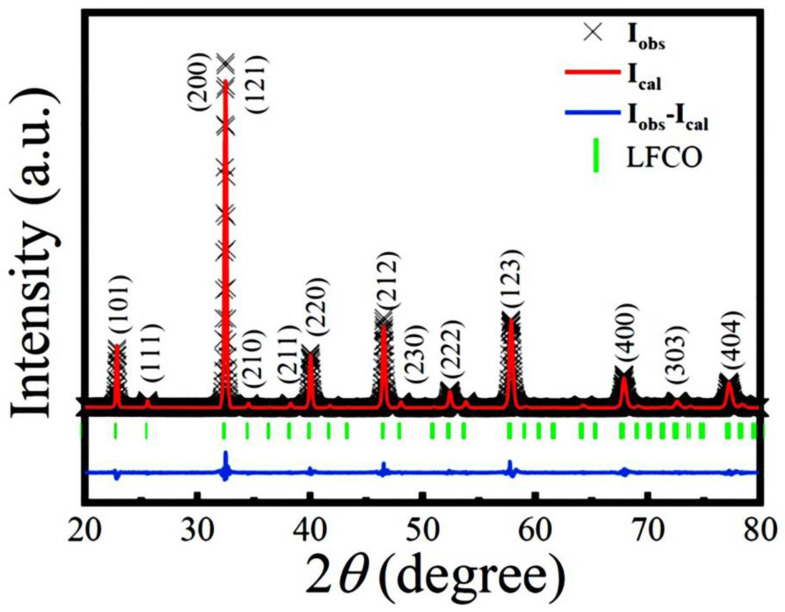
Comparison between the experimental XRD pattern and Rietveld refined profile for the LFCO powders at RT. The experimental data are denoted by cross marks, and the Rietveld refined profile is represented by a red solid line. Vertical green sticks mark the positions of allowed Bragg reflections.

**Figure 2 nanomaterials-13-03132-f002:**
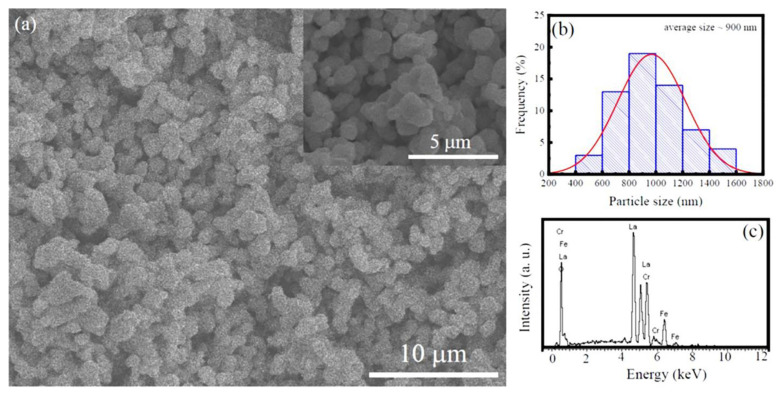
(**a**) Low-magnification SEM image of the LFCO powders. Inset is an enlarged SEM image. (**b**) Histogram of the LFCO particle size distribution. (**c**) EDS spectrum acquired in a mapping mode from the LFCO powders.

**Figure 3 nanomaterials-13-03132-f003:**
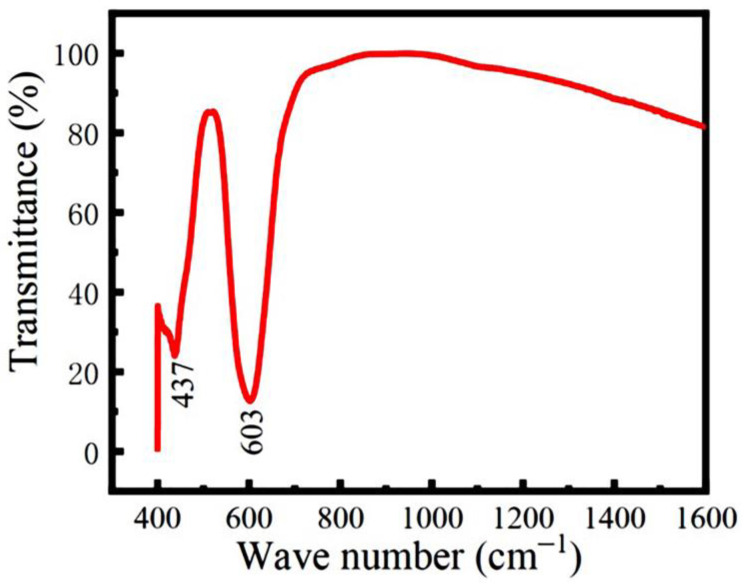
FTIR spectrum of the LFCO powders.

**Figure 4 nanomaterials-13-03132-f004:**
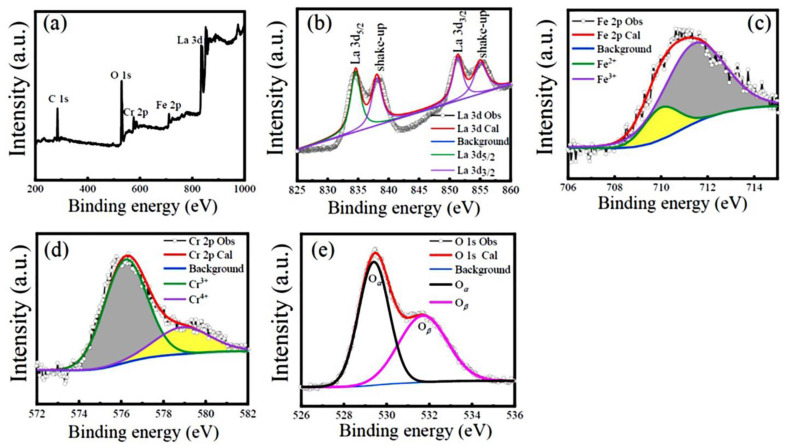
XPS spectra of (**a**) survey scan, and (**b**–**e**) high-resolution scan of La 3*d*, Fe 2*p*_3/2_, Cr 2*p*_3/2_, and O 1*s* for the LFCO powders.

**Figure 5 nanomaterials-13-03132-f005:**
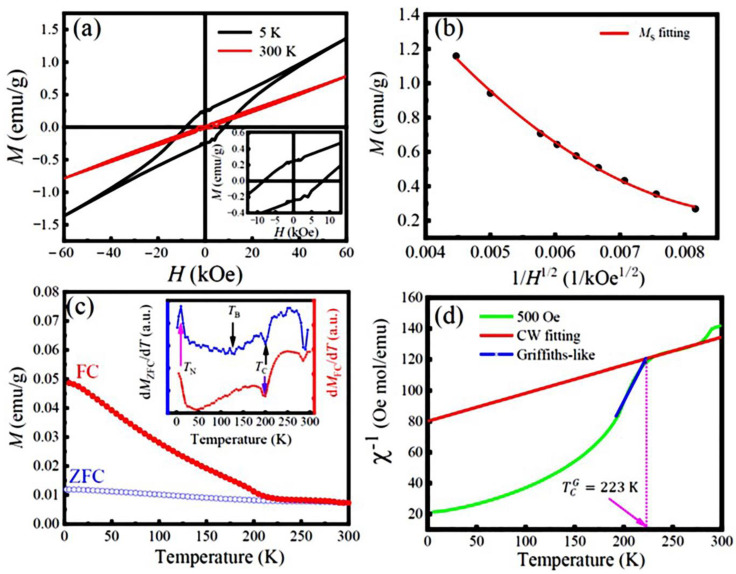
(**a**) *M*-*H* hysteresis loops of the LFCO powders measured at temperatures of 5 K and 300 K. Inset is the local *M*-*H* hysteresis loop recorded at 5 K. (**b**) Plot of *M(H)* vs. 1/$\sqrt{H}$ curve to extract the *M*_S_ value of the powders by the law of approach to saturation. (**c**) Magnetizations (*M*) of the LFCO powders measured as a function of temperature measured in ZFC and FC protocols, and (**d**) magnetic inverse susceptibilities (*χ*^−1^) with respect to the temperature under a magnetic field of 500 Oe. Inset in (**c**) represents the plots of d*M*_ZFC_/d*T* vs. *T* and d*M*_FC_/d*T* vs. *T*.

**Figure 6 nanomaterials-13-03132-f006:**
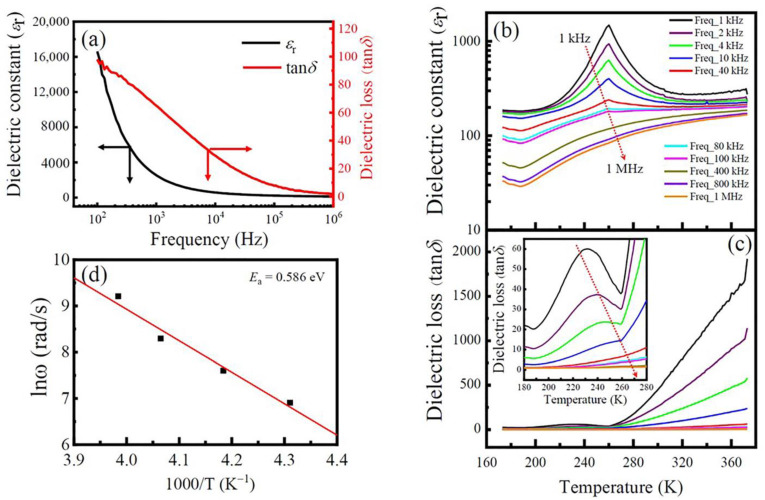
(**a**) Frequency dependence of dielectric constant (*ε*_r_) and dielectric loss (tan*δ*) of the LFCO ceramics measured at RT. (**b**,**c**) Temperature-dependent *ε*_r_ and tan*δ* of the LFCO ceramics measured a series of frequencies. Arrows in (**b**,**c**) show the direction of the frequency increasing. Inset in (**c**) represents tan*δ*–*T* curves between 180 K and 280 K. (**d**) Plot of Ln*ω* vs. 1000/*T* for the LFCO ceramics at low-temperature region.

**Figure 7 nanomaterials-13-03132-f007:**
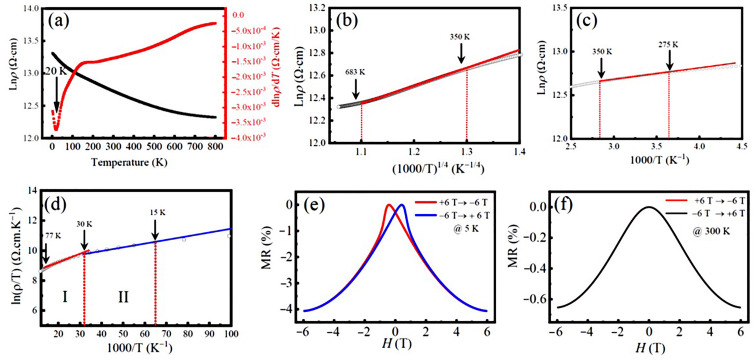
(**a**) The resistivity, Ln*ρ*(*T*), of the LFCO ceramics measured as a function of temperature from 2 K to 800 K without applying magnetic field, and the plot of d*ρ*/d*T* as a function of the temperature. (**b**) Ln(*ρ*) as a function of (1000/*T*)^1/4^ between 350 and 683 K. (**c**) Ln(*ρ*) as a function of 1000/*T* between 275 K and 350 K. (**d**) Ln(*ρ/T*) as a function of 1000/*T* between 15 K and 77 K, displaying two linear fitting regions, I (30–77 K) and II (15–30 K), respectively. Solid lines denote the linear fittings to the experimental data. (**e**,**f**) *MR*-*H* curves of the LFCO ceramics recorded at 5 K and 300 K, respectively.

## Data Availability

Data will be made available upon request to the corresponding author.

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
