# Peer review of "Microstructural Characterization and Magnetic, Dielectric, and Transport Properties of Hydrothermal La2FeCrO6 Double Perovskites"

_nanomaterials, 2023, doi:10.3390/nano13243132_

Round 1

Reviewer 1 Report

Comments and Suggestions for Authors

Authors have focused the “hydrothermal synthesis, structural characterization, magnetic, dielectric and transport properties of La2FeCrO6 double perovskites”. Authors need to make the Major revision work for publication.

1.     Page 1, line 29: Need to check the language errors.

2.     FTIR profile peaks need to be explained.

3.     All the XPS profiles need to be deconvoluted and highlighted.

4.     The magnetic properties discussions need to be explained more in detail.

5.     Conclusion section needs to be trimmed.

6.     English of the manuscript needs to be improved.

Comments on the Quality of English Language

English of the manuscript needs to be improved.

Reviewer 2 Report

Comments and Suggestions for Authors

1.     In the Introduction they should explain the importance of the perovskite nanostructures.

2.     The authors should explain the novelty of the material.

3.     The authors should compare the current results with similar reported studies.

4.     The synthesis process should be depicted by the diagram.  

5.     The authors should provide the importance of the hydrothermal method which is used to synthesize the required materials.

6.     The SEM studies should be improved. Elemental mapping should be added.

7.     The conclusion should be more concise. 

Comments on the Quality of English Language

Minor editing of English language required

Round 2

Reviewer 1 Report

Comments and Suggestions for Authors

Suitable for publication.